# Persistent Spinal Pain Syndrome Type 2 (PSPS-T2), a Social Pain? Advocacy for a Social Gradient of Health Approach to Chronic Pain

**DOI:** 10.3390/jcm10132817

**Published:** 2021-06-25

**Authors:** Nicolas Naiditch, Maxime Billot, Maarten Moens, Lisa Goudman, Philippe Cornet, David Le Breton, Manuel Roulaud, Amine Ounajim, Philippe Page, Bertille Lorgeoux, Kevin Nivole, Pierre Pries, Cecile Swennen, Simon Teyssedou, Elodie Charrier, Géraldine Brumauld de Montgazon, Pierre François Descoins, Brigitte Roy-Moreau, Nelly Grimaud, Romain David, Tanguy Vendeuvre, Philippe Rigoard

**Affiliations:** 1PRISMATICS Lab (Predictive Research in Spine/Neuromodulation Management and Thoracic Innovation/Cardiac Surgery), Poitiers University Hospital, 86021 Poitiers, France; maxime.billot@chu-poitiers.fr (M.B.); Manuel.ROULAUD@chu-poitiers.fr (M.R.); Amine.OUNAJIM@chu-poitiers.fr (A.O.); Bertille.LORGEOUX@chu-poitiers.fr (B.L.); Kevin.NIVOLE@chu-poitiers.fr (K.N.); romain-david@hotmail.fr (R.D.); t.vendeuvre@gmail.com (T.V.); Philippe.RIGOARD@chu-poitiers.fr (P.R.); 2EURIDOL, Neuropôle de Strasbourg, Faculty of Life Science, University of Strasbourg, 67000 Strasbourg, France; 3Dyname, UMR 7367, Faculty of Social Sciences, University of Strasbourg, 67083 Strasbourg, France; dav.le.breton@orange.fr; 4Department of Neurosurgery, Universitair Ziekenhuis Brussel, 1090 Brussels, Belgium; mtmoens@gmail.com (M.M.); lisa.goudman@gmail.com (L.G.); 5Department of General Medicine, Sorbonne University, 75012 Paris, France; cornetphilippe@hotmail.com; 6Department of Spine Surgery & Neuromodulation, Poitiers University Hospital, 86021 Poitiers, France; Philippe.PAGE@chu-poitiers.fr (P.P.); Pierre.PRIES@chu-poitiers.fr (P.P.); Cecile.SWENNEN@chu-poitiers.fr (C.S.); Simon.TEYSSEDOU@chu-poitiers.fr (S.T.); 7Pain Evaluation and Treatment Centre, Poitiers University Hospital, 86021 Poitiers, France; Elodie.CHARRIER@chu-poitiers.fr; 8Pain Evaluation and Treatment Centre, La Rochelle Hospital, 17000 La Rochelle, France; Geraldine.DEMONTGAZON@ght-atlantique17.fr; 9Pain Evaluation and Treatment Centre, Nord Deux-Sèvres Hospital, 79000 Niort, France; pierre-francois.descoins@ch-niort.fr (P.F.D.); Roy-Moreau.Brigitte@chnds.fr (B.R.-M.); 10Pain Evaluation and Treatment Centre, Centre Clinical Elsan, 16800 Soyaux, France; ngrimaud@centre-clinical.fr; 11Physical and Rehabilitation Medicine Unit, Poitiers University Hospital, University of Poitiers, 86021 Poitiers, France; 12Pprime Institute UPR 3346, CNRS, ISAE-ENSMA, University of Poitiers, 86360 Chasseneuil-du-Poitou, France; 13Department of Orthopaedic Surgery and Traumatology, Centre Hospitalier Universitaire de Poitiers, 86021 Poitiers, France; 14ABS Lab, Poitiers University, 86021 Poitiers, France

**Keywords:** social gradient of health, persistent spinal pain syndrome type 2, failed back surgery syndrome, social determinants of health

## Abstract

The Social Gradient of Health (SGH), or position in the social hierarchy, is one of the major determinants of health. It influences the development and evolution of many chronic diseases. Chronic pain dramatically affects individual and social condition. Its medico-economic impact is significant and worldwide. Failed Back Surgery Syndrome or Persistent Spinal Pain Syndrome type 2 (PSPS-T2) represents one of its most fascinating and disabling conditions. However, the influence of SGH on PSPS-T2 has been poorly explored. We designed a prospective multicentric study (PREDIBACK study) to assess the SGH prevalence, and to examine its association with medical and psychological variables, in PSPS-T2 patients. This study included 200 patients to determine the SGH association with pain (NPRS), Quality of life (EQ-5D-5L), kinesiophobia (FABQ-Work), catastrophism (CSQ), and functional capacity (ODI). Around 85.3% of PSPS-T2 patients in our study had low SGH. Low SGH patients had a higher FABQ-Work and CSQ-Catastrophizing score than high SGH patients (*p* < 0.05). High SGH patients have a higher ODI score than low SGH patients (*p* < 0.10). Our results suggest that SGH is a relevant factor to guide prevention, research, and ultimately intervention in PSPS-T2 patients and could be more widely transposed to chronic pain.

## 1. Introduction

Between 10% and 50% of patients who have undergone lumbar spinal surgery still experience the intense persistent pain and impaired function known as Failed Back Surgery Syndrome (FBSS), which has recently been proposed as Persistent Spinal Pain Syndrome type 2 (PSPS-T2) [1,2,3]. PSPS-T2 diagnosis is related to an illness trajectory. The concept refers to the course of an illness and to the entire organization of the medical work carried out to follow this course, i.e., the care pathway [4]. For most of these patients, the illness trajectory starts with acute Low Back Pain (LBP) management failure (pain duration <3 months), which continues with chronic LBP management failure that can lead to lumbar spinal surgery. Unfortunately, even after anatomical and radiological successful outcomes, this spine surgery may not relieve LBP and/or can result in the development of post-operative chronic pain [1]. While PSPS-T2 may affect the mainstream population, and has been considered biologically heterogeneous [5], its development and evolution may also be influenced by psychological and social factors [6,7,8].

Over the last few decades, research has, nonetheless, mainly been limited to examination of biological factors, thereby neglecting other factors [9]. While recent publications have observed that social factors are relevant to the patient care pathway, they remain rarely documented [10]. In the literature, however, social factors have been shown to affect chronic pain diseases [6,11], particularly throughout the Social Gradient of Health (SGH) examination.

The Social Gradient of Health (SGH) is a concept used to describe the relationship between the socioeconomic position and health [12]. It has been reported that people with a low socioeconomic position have worse physical and mental health than people with a high socioeconomic position [11,12]. There is consistent evidence of a higher level of physical activity and sport practice in the general population with high SGH compared to people with low SGH [13,14]. Furthermore, patients with low social status, low education, and low incomes (low SGH), present a low level of health literacy (i.e., low capacities to obtain, communicate, process, and understand basic health information and services allowing them to make appropriate health decisions) [15]. Tobacco use and alcohol consumption are also more widespread in people with low SGH [16]. While PSPS-T2 might be influenced by SGH, there is no available evidence in the literature. Moreover, the term “psychosocial”, commonly used in the literature, fosters confusion between the psychological and social dimensions [17]. In this context, the influence of the social dimension on PSPS-T2, and, more specifically, the SGH, should be separately investigated.

The main objective of this study was to determine the association between SGH and PSPS-T2 prevalence in 200 patients consulting a pain specialist. Furthermore, we evaluated SGH association, with the main medical and psychological pain assessment tools, among 200 patients included consecutively in a prospective, multicentric observational study. Our results could be used to integrate SGH in the evaluation/diagnosis, orientation and treatment of PSPS-T2 patients, and ultimately to optimize the medical care pathway.

## 2. Materials and Methods

### 2.1. Study Design

Two hundred PSPS-T2 patients included in the prospective, multicenter, observational PREDIBACK study were considered for our study. The primary objective of the PREDIBACK study was to clinically, psychologically, and socially characterize PSPS-T2 patients (https://clinicaltrials.gov/ct2/show/NCT02964130; First Posted: 15 November 2016). Patients were recruited consecutively and monitored for 12 months in 5 pain clinics in the New-Aquitaine Region: Angoulême, Bressuire, La Rochelle, Niort, and Poitiers (France). Patient recruitment started in January 2017 and was completed in March 2018. The study was approved by the ANSM (2016-A01144-47) and the Ethics Committee (CPP Ouest III).

### 2.2. Patient Selection

Inclusion criteria: PSPS-T2 patients were identified at each site through standard clinical practice. To be eligible, patients had at least one spinal surgery, post-operative leg and/or LBP for at least six months, and an average global pain score greater than, or equal to, 4 on the Numeric Pain Rating Scale (NPRS) [18]. All the patients gave their informed consent before enrolment.

Non-inclusion criteria: Patient is, or has been, treated with Spinal Cord Stimulation, subcutaneous or peripheral nerve stimulation, an intrathecal drug delivery system; has life expectancy of less than 12 months beyond study enrollment; Patient is unable to undergo study assessments or to complete questionnaires independently; is a member of a vulnerable population; or investigator suspects substance abuse that might confound the study results.

### 2.3. Measures

All measurements were collected during the PREDIBACK Study inclusion visit.

The social gradient was measured using the Profession and Socioprofessional Category (PSC). PSC is a statistical nomenclature used to classify occupation [19]. PSC was assessed from the patients’ profession and coded by a sociologist according to the French National Institute of Statistics and Economic Studies nomenclature [20]. In retired and unemployed patients, their last job was used for the analysis [21].

The choice of SGH indicators among patients with chronic LBP is important [22]. The French High Council for Public Health recommends measuring the SGH using not only Profession and Socioprofessional Category but also educational attainment and employment status and incomes [23]. Among these factors, we did not have information on income, and it appeared, to us, that professional status was largely affected by the pain. We preferred PCS to educational level because it is more relevant in France to evaluate SGH and to thereby consider possible social mobility.

Educational level was collected through a qualitative ordinal variable proposing the different French educational levels. It was recoded in two modalities. Early childhood education; primary education; lower secondary education and vocational were grouped together in a new modality called “<upper secondary education”. Upper secondary education; short-cycle tertiary education; bachelor or equivalent level; master or equivalent level and doctoral or equivalent level were grouped as “≥upper secondary education”. We have chosen the upper secondary education as cut-off for its theoretical relevance in the Social Gradient of Health study.

The professional situation of patients was collected on a declarative basis according to 5 modalities: active, disabled, long-term sick leave, sick leave, and active unemployed; it was then recoded. Patients on disability, long-term sick leave, sick leave, and unemployment were grouped together in a new modality called “inactive”. Analysis focused on normally active and working-age patients. Retired and housewives were excluded from the analysis because their inactivity could be considered normal.

Global pain intensity over the past 5 days was assessed by the Numeric Pain Rating Scale (NPRS) [24]. A perceived score of 0 represents a total absence of pain and 10 represents the worst pain imaginable.

Functional disability was assessed by the 10 items of the Oswestry Disability Index (ODI) [25]. A perceived score of 0 represents no disability and 50 represents total disability.

Quality of life was assessed by the 5 items of the EuroQol 5-Dimensional 5-Level questionnaire (EQ-5D-5L) [26]. Health state index scores generally range from less than 0 (where 0 is the value of a health state equivalent to dead; negative values representing values as worse than dead) to 1 (the value of full health)

Depressive syndromes were measured by the 14 items of the Hospital Anxiety and Depression Scale (HADS) [27]. This questionnaire consists of two subscales. The 7 items of HADS-anxiety (HADS-A) measure anxiety, and the 7 items of HADS-depression (HADS-D) measure depression. Each subscale has a minimum score of 0 and a maximum score of 21. The higher the score, the greater the probability of suffering from depressive and/or anxiety symptoms.

Kinesiophobia is defined as pathological fear of motion and is assessed by the 16 items of the Fear-Avoidance Belief Questionnaire (FABQ) [28,29]. This questionnaire consists of two subscales. The 5 items of the FABQ-Physical Activity (FABQ-PA) subscale measure kinesiophobia associated with physical activity, and the 11 items of FABQ-Work (FABQ-W) subscale measure kinesiophobia associated with work. Each subscale has a minimum score of 0. The maximum score for FABQ-PA is 24 and for FABQ-W is 42. Level of kinesiophobia is positively correlated with the FABQ score.

Cognitive strategies for coping with pain were evaluated with the French adaptation of the Coping Strategies Questionnaire (CSQ) [30,31]. It consists of six subscales for cognitive strategies including pain ignorance (5 items), reinterpretation (4 items), diversion (5 items), self-encouragement (4 items), catastrophizing (6 items), and praying/hoping (3 items). The praying subscale was not used in this study because it is considered sensitive data. Each item consists of a Likert scale ranging from 1 “never” to 4 “always”, indicating how frequently the strategy is used to cope with pain.

Having already consulted a psychologist or not from the outset of the illness trajectory was recorded on a declarative basis. It was recoded in two ways: “No, I have never consulted a psychologist” and “Yes, I have consulted a psychologist”.

The representation of the psychologist’s usefulness in pain management was assessed with an 11-point Likert scale (0 = Not at all useful; 10 = Quite useful). Patients who had never met a psychologist were allowed to answer this question.

### 2.4. Statistical Analysis

The quantitative variables were described by their mean and standard deviation (SD). Qualitative variables were described by frequency and percentages. Normality of the data distribution was assessed by the Shapiro-Wilk test.

In order to evaluate the influence of the SGH on the prevalence of PSPS-T2 patients consulting within a pain structure, we compared the PSC of the PREDIBACK study with the regional PSC of the same age group (extracted from the statistics of the French National Institute of Statistics and Economic Studies). We used a multinomial goodness of fit test to measure distributional differences.

Farmers (PSC 1), Craftsmen, salesmen and managers (PSC 2), Blue-collar workers (PSC 6), and Lower-grade white-collar workers (PSC 5) were grouped as “Low Social Gradient of Health (SGH−)”. Technicians, associate professionals (PSC 4), and Professionals (PSC 3) were grouped as “High Social Gradient of Health (SGH+)”.

The relationship between SGH and educational level, gender, professional situation, and having already consulted a psychologist were analyzed together with the Chi² or Fisher test depending on the number of people in the different modalities. Relationships between the SGH and NPRS, ODI, EQ-5D-5L, HAD, CSQ, FABQ, and the representation of the psychologist’s usefulness were determined using the Student or Mann-Whitney-Wilcoxon tests, depending on the normality of the quantitative variable.

To assess the effect of SGH on each factor, we performed a multivariate analysis using logistic regression. Bivariate analyses were conducted to identify variables that could be included in the models (variables with a *p*-value ≤ 0.05). Variables statistically insignificant but theoretically relevant, were included in the models. A step-by-step selection of statistically significant variables was conducted.

The bivariate and multivariate analyses were conducted as available-case analyses based on completed assessments (i.e., deleting a case when it is missing a variable required for a particular analysis, but including that case in analyses for which all required variables are present).

Statistical analyses were performed with the R software (R Development Core Team, 2010), and two-tailed *p* values of <0.05 were considered to be significant.

## 3. Results

### 3.1. Participants

From the 200 patients, 9 patients were withdrawn from the study because the information provided did not make it possible to assess their PSC. Our final sample consisted of 191 patients with an average age of 52.8 years (12.5). Females were 55.5% (106/191) and males 44.5% (85/191).

### 3.2. PSC Representation in the PREDIBACK Study and Comparison with the Regional Population

Results of the PSC representation, and comparison with the regional population, are presented in Table 1.

Farmers represented 1.6% (3/191) of the PSC structure of the PREDIBACK study. Craftsmen, salesmen and managers likewise represented 1.6% (3/191), blue-collar workers 33.5% (64/191), lower-grade white-collar workers 48.7% (93/191), technicians, associate professionals 9.9% (19/191), and professionals 4.7% (9/191).

The PSC structure of the PREDIBACK study differs significantly from the regional population (*p* < 0.01). Farmers (−1.9), craftsmen, salesmen and managers (−7.5), technicians, associate professionals (−13.7), and professionals (−9.2) were underrepresented in the PREDIBACK study compared to the regional population. Blue-collar workers (+12.4) and lower-grade white-collar workers (+19.7) were overrepresented in the PREDIBACK study compared to the regional population. Patients with a Low Social Gradient of Health (SGH−) were significantly overrepresented (+22.5) and patients with a High Social Gradient of Health (SGH+) were significantly underrepresented (−22.6) in the PREDIBACK study compared to the regional population (*p* < 0.01).

### 3.3. Social Characteristics of the Patients According to SGH Group

Results of the social characteristics of the patients are presented in Table 2.

The average age was not significantly different, according to SGH (*p* = 0.080). Women were significantly more numerous among SGH+ patients than men (*p* = 0.008). SGH− patients had a significantly lower educational level than SGH+ patients (*p* < 0.001). SGH− patients were without professional activity significantly more than SGH+ patients (*p* = 0.002).

### 3.4. Association between SGH Group and Medical Assessment Tools

Results of the SGH influence on clinical factors are presented in the Table 3.

The pain intensity (NPRS) and quality of life (EQ-5D-5L) scores were not significantly different between SGH+ and SGH− patients (*p* = 0.889 and *p* = 0.586, respectively). The mean functional disability (ODI) score was higher in SGH+ patients than in SGH− patients, but the difference was not significant (*p* = 0.071).

### 3.5. Association between SGH and Psychological Assessment Tools

Results of the association between SGH and psychological assessment tools are presented in Table 4.

The anxiety and depression scores (HAD) were not significantly different according to SGH (*p* = 0.227). Nor were significant differences found in pain coping strategies (CSQ) for distraction strategies (*p* = 0.273), reinterpretation (*p* = 0.414), ignorance (*p* = 0.819), or self-encouragement (*p* = 0.770). The CSQ catastrophizing score was higher in SGH− patients than in SGH+ patients, but the difference was not significant (*p* = 0.084).

The kinesiophobia associated with physical activity score (FABQ-PA) was not significantly different according to SGH (*p* = 0.436). However, the kinesiophobia, associated with work score (FABQ-W), was significantly higher in SGH− than in SGH+ patients (*p* = 0.043). The average rating for a psychologist’s usefulness was significantly lower in SGH− than in SGH+ patients (*p* = 0.002). The former were significantly more numerous to have never consulted a psychologist than the latter (*p* = 0.003).

### 3.6. Multivariate Analysis

Multivariate analysis results are presented in Table 5 and Figure 1.

Patients with high ODI scores were significantly more likely to be SGH+ than SGH− (coef. = 0.280; 95% CI [0.014; 0.547]; *p* = 0.039).

Patients with a high CSQ-Catastrophizing score were significantly more likely to be SGH− than SGH+ (coef. = −0.303; 95% CI [−0.585; −0.020]; *p* = 0.036). Patients with a high score of FABQ-W, associated with work activity, tended to be more SGH− than SGH+ (coef. = −0.214; 95% CI [−0.461; 0.034]; *p* = 0.091). Patients who had previously consulted a psychologist were significantly more SGH+ than SGH− (coef. = 0.299; 95% CI [0.055; 0.542]; *p* = 0.016).

In order to measure a potential gender moderation effect related to the over-representation of women in SGH+ patients, we included it in a new regression model. Women were more likely to be SGH+ than SGH− (coef. = 0.349; IC95% [0.070; 0.627;]; *p* = 0.014]). Gender did not have a significant moderating effect on the significance of the ODI score (*p* = 0.033), FABQ-W score (*p* = 0.078), CSQ-C (*p* = 0.046), or psychologist consultation (*p* = 0.030).

The model parameters are: *n* = 177; R2 = 0.181; *p*-value = 0.001. Variables colored in green are statistically significant (*p* < 0.05). Variable colored in pink is not statistically significant (*p* = 0.091). ODI: Oswestry Disability Index; CSQ: Coping Strategies Questionnaire; FABQ-W: Fear-Avoidance Belief Questionnaire, Work subscale.

## 4. Discussion

The main objective of this study was to determine the association between the Social Gradient of Health (SGH) and Persistent Spinal Pain Syndrome Type 2 (PSPS-T2) prevalence in patients consulting a pain specialist. Results showed that patients with Low SGH (SGH−) were over-represented and that patients with High SGH (SGH+) were under-represented. Furthermore, patients with high CSQ-Catastrophizing (in multivariate analysis) score and FABQ-Work score (univariate analysis) were significantly more likely to be SGH− than SGH+.

### 4.1. The Need for an SGH Approach to Stratify the PSPS-T2 Population

Compared to the regional data, the analysis of the PSC showed that blue-collar workers and lower-grade white-collar workers were overrepresented (+12.4 and +19.7, respectively), and technicians, associate professionals, and professionals were underrepresented (−13.4 and −9.2, respectively) in a population of PSPS-T2 patients. Although farmers and craftsmen, salesmen, and managers have low socioeconomic status, they were surprisingly underrepresented (−1.9 vs. −7.5). Based on a sample of 26,500 people, representative of the National French population, the health and social protection survey highlighted lower medical specialist consultation rates among craftsmen, salesmen and managers, blue-collar workers and farmers than for managers [32]. According to the authors, the differences were mainly due to healthcare renunciation for economic reasons and transportation difficulties. Furthermore, in accordance with the traditional model of masculinity, particularly present among SGH−, men traditionally have a more physically demanding work. In this context, physical effort and suffering/pain are valued in a productive work environment. The lower use of medical specialist consultation in SGH− patients may thus be the expression of their relation to their bodies and pain perception. All in all, PSPS-T2 patients could leave the health care system without any pain relief. Integrating SGH in their pain assessment and follow-up would make it possible to adapt care to the patient social characteristics and to avoid the chronification process and medical wandering.

### 4.2. Impact of SGH on PSPS-T2 Prevalence

Similarly to the greater prevalence of SGH− than SGH+ in chronic LBP patients [33,34,35,36], our results indicated that SGH− PSPS-T2 patients were overrepresented in comparison with regional data (+22.6 points, *p* < 0.01). Besides, Plouvier et al. [34] reported negative association between SGH and the duration of exposure to biomechanical strains in 1487 workers with persistent, or recurrent, LBP. They concluded that biomechanical factors, related to physical exposure at work, were the main influencing factor in the relation between LBP and SGH. In addition, SGH− reported higher incidence of LBP and were more likely to develop pain chronification than SGH+ [22,37,38]. In their review based on 66 articles, Dionne et al. [37] suggested that SGH may be considered as a “marker” for other factors involved in the etiology of the disease. The fact that LBP was related to SGH was supported by five non-exclusive hypotheses: (i) behavioral and environmental risk factors, (ii) occupational factors, (iii) compromised “health stock”, (iv) health service access and utilization, and (v) adaptation to stressful events [37]. Among 3150 French patients having undergone lumbar disc surgery, Fouquet et al. [39] revealed that SGH− were more exposed to excess risk at work and that 20% of lumbar disc surgery in men could theoretically be avoided with effective prevention programs for the SGH− population. In case of surgery for lumbar spinal stenosis in 13,406 patients, Iderberg et al. [40] showed that socioeconomic factors significantly affect pain and functional disability outcomes. The authors concluded that surgical complications are more numerous and clinical outcomes poorer for SGH− than for SGH+ patients. Authors reported that comorbidities such as smoking, alcohol consumption, obesity, social welfare, or unemployed strongly affected surgery outcomes in SGH− patients [40]. All in all, we can suggest that the cumulative effect of social inequalities results in greater risk of developing PSPS-T2 in SGH− (Figure 2). Using SGH as an early screening tool could help to stratify and to identify the most appropriate medical care pathway according to the social characteristics of the patient.

This figure illustrates the cumulative influence of the SGH on the theoretical proportion of patients according to their position in the socioeconomic hierarchy at each stage of the PSPS-T2 patients’ illness trajectory. The different shades of pink (from pink to red) indicate the new proportion of SGH− patient at each pain illness level, from acute to refractory PSPS-T2. This figure shows that (i) the proportion of SGH+ decrease while the pain illness level increase (from acute pain to PSPS-T2); (ii) the proportion of SGH− increase while the pain illness level increases. At each stage of the illness trajectory, the proportion of SGH− patients is increasingly important. © PRISMATICS Lab. PSPS-T2: Persistent Spinal Pain Syndrome Type 2

### 4.3. Impact of SGH on ODI, FABQ-Work and CSQ-Catastrophizing

The literature has shown that functional disability is related to kinesiophobia and pain catastrophizing [41,42,43]. Thomas et al. [42] found that functional disability was positively correlated with catastrophizing and kinesiophobia among 50 patients with chronic LBP. By using a population-based cohort of the general Dutch population (*n* = 1571), Picavet et al. [41] reported that the highest tertile of pain catastrophizing, or kinesiophobia, increases by more than three times the risk of developing chronic LBP and disability. According to these findings, we expected that patients with the highest score of functional disability, kinesiophobia, and pain catastrophizing would present the same social characteristics. However, our results revealed that SGH− had lower functional disability scores (ODI: coef. = 0.280; *p* = 0.039), higher kinesophobia (FABQ-W: coef. = −0.214; *p* = 0.091), and catastrophizing scores (CSQ-Catastrophizing: coef. = −0.303; *p* = 0.036) than SGH+ patients. In their recent publication, Goudman et al. [44] reported no significant correlation between subjective and objective functional disability assessed respectively with the ODI and an accelerometer device in 39 PSPS-T2 patients. Since ODI measures subjective functional disability, it could be hypothesized that SGH− have less functional disability perception than SGH+ patients. This hypothesis corroborates the virile body representation of the SGH− population, notably marked by a form of stoicism manifesting “a self-defense” [45]. While changes in the labor market have led to reduced social differences, the traditional value of fatigue resistance is still topical [46] and may support the lower ODI scores in SGH− patients. Otherwise, in 1846 workers, aged 20–64 years, Hämmig and Baueur [47] reported that SGH− were more affected by physical workloads (carrying heavy loads, painful or tiring posture, etc.) and that SGH+ were more affected by psychosocial work demands and job resources (high time pressure, steadily growing workload, etc.) Although functional disability appeared lower in the current study for SGH− PSPS-T2 patients, the greater physical workloads may contribute to the higher score for work-related kinesiophobia (FABQ-W) [29,48]. Similarly, higher pain catastrophism has been reported in SGH− with LBP or total knee arthroplasty [43,49]. Authors suggested that time to care access, physician bias, and patient education may contribute to the higher catastrophism observed in SGH− patients [50]. All in all, our findings associated with the literature demonstrate that SGH− patients, especially those with PSPS-T2, must be treated with specific care by carefully considering the social components in clinical and psychological outcomes. For that very reason, an innovative model of multidisciplinary holistic therapeutic care should be considered.

### 4.4. Innovative Model of Multidisciplinary Holistic Therapeutical Care

Our results showed that (i) PSPS-T2 patients were characterized by homogeneous social profiles (regardless of biological heterogeneity) and that (ii) social factors substantially influenced psychological and functional outcomes. The critical extent of social factors highlighted by the current study can be used to propose the following innovative model for a multidisciplinary holistic therapeutic care pathway:Systematic multidisciplinary consultation with the necessary presence of a social worker to optimize the medical care pathway.Use of SGH as a weighting tool to optimize clinical, functional disability, and psychological outcome evaluation. SGH can be used in the same way as educational level is used to adjust the threshold of cognitive impairment evaluation with the Mini-Mental State Examination [51,52].Use of SGH as a stratification tool that enhances and orients toward an optimized medical care pathway, notably by providing personalized information and support through patient education programs.

Thus, the main goal of this model is to continue effort to break down social barriers in close cooperation with medical professionals. This model will provide new opportunities to improve the medical care pathway of PSPS-T2 patients and can be easily transposable to other chronic pain diseases.

### 4.5. Study Strengths and Limitations

Even if our study is a prospective, multicentric observational study performed with a large sample of 200 patients with PSPS-T2, several limitations should be considered. First, all the patients included in the study were previously involved in a specific pain consultation and might not constitute a representative sample of PSPS-T2 patients in the general population. However, our sample of 200 patients was large and results were congruent with the huge worldwide literature on LBP. Our findings can provide a real springboard for the consideration of social factors, and more specifically SGH, in clinical and psychological evaluation of PSPS-T2 patients and other chronic pathologies as well. Furthermore, our study was focused on SGH, whereas several other factors (gender, professional status...) could also influence clinical outcomes. Finally, our study could be used to define a starting point to building predictive model of therapy responses throughout the 12-month follow-up, and might be refined by further research.

## 5. Conclusions

Our prospective, multicentric observational study highlighted that SGH− were overrepresented and constitute a large majority of patients with PSPS-T2 compared to the general socio-economic structure. Our study also emphasizes an association between SGH and main functional (ODI score) and psychological (kinesiophobia, catastrophizing) pain assessment tools. It confirms that social factors should be considered at fair value and above all, as in other chronic illnesses, must not be neglected in PSPS-T2 patients. This study lays the groundwork for an innovative model of multidisciplinary holistic therapeutic care to improve prevention strategies and management of the medical care pathway, and it could thereby serve as a substrate of future research.

## Figures and Tables

**Figure 1 jcm-10-02817-f001:**
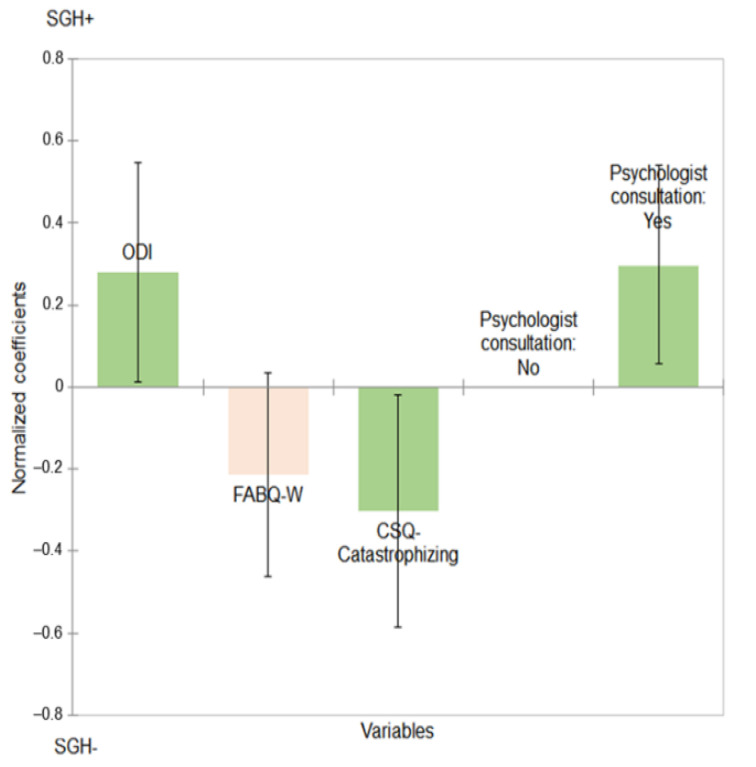
Representation of the logistic regression showing the standardized coefficients (95% CI) between Social Gradient of Health (SGH) and several dependent variables.

**Figure 2 jcm-10-02817-f002:**
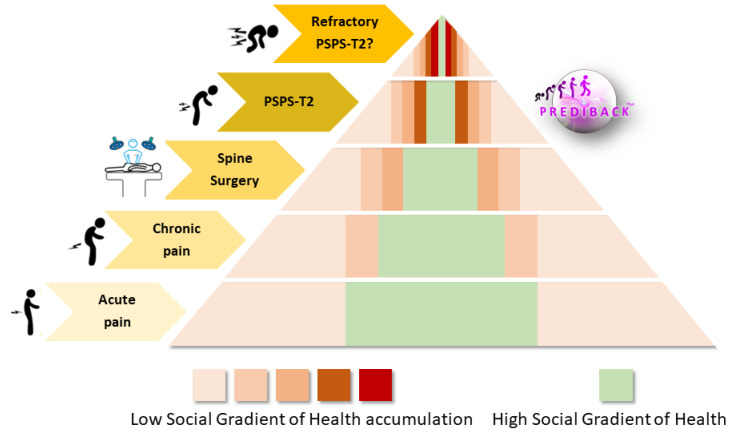
Illustration of the theoretical evolution of the proportion of patients with low and high Social Gradient of Health according to the different stages of the illness trajectory of Persistent Spinal Pain Syndrome type 2 (PSPS-T2) patients.

**Table 1 jcm-10-02817-t001:** Comparison between occupational categories of patients in the PREDIBACK study and the regional population aged between 40 and 64 years in 2015.

Professions and Socio-Professional Categories	PREDIBACK Population		Regional Population		Difference between %
	*n* = 191	%	*n* = 1,492,365	%	
**Patients SGH−**	163	85.3	93,693	62.8	+22.5
Farmers (PSC 1)	3	1.6	51,820	3.5	−1.9
Craftsmen, salesmen and managers (PSC 2)	3	1.6	136,378	9.1	−7.5
Blue-collar workers (PSC 6)	64	33.5	315,381	21.1	+12.4
Lower-grade white-collar workers (PSC 5)	93	48.7	433,214	29.0	+19.7
**Patients SGH+**	28	14.6	555,572	37.2	−22.6
Technicians, associate professionals (PSC 4)	19	9.9	348,087	23.3	−13.4
Professionals (PSC 3)	9	4.7	207,485	13.9	−9.2

Difference between the PSC structure of the PREDIBACK study and the regional population, *p*-value < 0.01. SGH−: Low Social Gradient of Health; SGH+: High Social Gradient of Health; PSC: Profession and Socioprofessional Category.

**Table 2 jcm-10-02817-t002:** Social characteristics of patients according to the Social Gradient of Health (SGH).

Variables	SGH−		SGH+		*p*-Value
	*n*	%	*n*	%	
Age (Mean; SD)	52.1	12.6	56.7	11.6	0.080
Gender					
Women	84	51.5	22	78.6	0.008
Men	79	48.5	6	21.5	
Educational level					
<upper secondary education	123	75.9	3	10.7	0.001
≥upper secondary education	39	24.1	25	89.3	
Professional situation					
Active	29	22.1	11	55.0	0.002
Inactive	102	77.9	9	45.0	

SGH−: Low Social Gradient of Health; SGH+: High Social Gradient of Health.

**Table 3 jcm-10-02817-t003:** Comparison of patients’ key medical assessment tools according to the Social Gradient of Health (SGH).

Variables	SGH−		SGH+		*p*-Value
	Mean	SD	Mean	SD	
NPRS	6.0	1.5	6.1	1.4	0.889
EQ-5D-5L	0.267	0.256	0.239	0.254	0.586
ODI	43.9	13.9	49.2	15.0	0.071

SGH−: Low Social Gradient of Health; SGH+: High Social Gradient of Health; NPRS: Numeric Pain Rating Scale; EQ-5D-5L: EuroQol 5-Dimensional 5-Level; ODI: Oswestry Disability Index.

**Table 4 jcm-10-02817-t004:** Comparison of patients’ psychological characteristics according to the Social Gradient of Health (SGH).

Variables	SGH−		SGH+		*p*-Value
	Mean	SD	Mean	SD	
HAD-D	8.7	4.2	8.1	3.0	0.701
HAS-A	10.3	4.1	9.4	3.8	0.785
CSQ-Distraction	11.8	4.0	12.7	4.2	0.273
CSQ-Reinterpretation	6.3	2.3	6.3	3.3	0.414
CSQ-Catastrophizing	9.9	3.0	8.9	2.7	0.084
CSQ-Ignorance	9.5	3.9	9.4	3.3	0.819
CSQ-Self Encouragement	10.4	2.7	10.7	2.9	0.770
FABQ-W	18.4	11.1	13.8	11.5	0.043
FABQ-PA	16.4	7.0	16.2	5.0	0.436
Representation of the psychologist’s usefulness	3.7	3.6	6.0	3.6	0.002
Psychologist consultation					
No, never (*n*; %)	98	65.8	10	35.7	0.003
Yes, at least once (*n*; %)	51	34.2	18	64.3	

HAD-D: Hospital Anxiety and Depression Scale, Depression subscale;HAD-A: Hospital Anxiety and Depression Scale, Anxiety subscale; CSQ: Coping Strategies Questionnaire; FABQ-W: Fear-Avoidance Belief Questionnaire, Work subscale; FABQ-PA: Fear-Avoidance Belief Questionnaire, Physical Activity subscale.

**Table 5 jcm-10-02817-t005:** Logistic regression describing the relationship between the Social Gradient of Health (SGH) and several dependent variables.

Variables	Normalized Coefficients *	CI95%.	*p*-Value
ODI	0.280	[0.014; 0.547]	0.039
CSQ-Catastrophizing	−0.303	[−0.585; −0.020]	0.036
FABQ-W	−0.214	[−0.461; 0.034]	0.091
No, I have never consulted a psychologist	Reference	Reference	0.016
Yes, I have consulted a psychologist	0.299	[0.055; 0.542]	

The model parameters are: *n* = 177; R^2^ = 0.181; *p*-value = 0.001; * Patients SGH− (*n* = 149) are represented by 0 and SGH+ (*n* = 28) by 1 in the independent variable. ODI: Oswestry Disability Index; CSQ: Coping Strategies Questionnaire; FABQ-W: Fear-Avoidance Belief Questionnaire, Work subscale.

## Data Availability

Not applicable.

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
