# Peer review of "Persistent Spinal Pain Syndrome Type 2 (PSPS-T2), a Social Pain? Advocacy for a Social Gradient of Health Approach to Chronic Pain"

_jcm, 2021, doi:10.3390/jcm10132817_

Round 1
Reviewer 1 Report
This is a well performed and scientifically stringent study showing a relationship between socioeconomic position and health status in patients with persitent low back and/or leg pain and impaired functional level who have undergone previous spinal surgery. The statistical analysis is solid and the authors conclude that the model that they present using The Social Gradient of Health (SGH) concept may provide new opportunities to improve the medical care pathway of patients with refractory pain after spine surgery.
I congratulate the authors for performing this study on a topic that has been intensely debated recently!! I have no major objections but there are some minor points that the authors need to consider and revise accordingly:
- Persistent Spinal Pain Syndrome type 2 is NOT a generally accepted term to replace the acronym FBSS (Failed Back Surgery Syndrome). It is still a suggestion for replacement which has not been widely adopted by the scientific community dealing with this condition! Most surgeons and pain clinicians still use the term FBSS and it is accordingly NOT appropriate to state "previously named" in the Abstract (page 1, line 40-41) or "recently rebaptized" in the Introduction (page 2, line 58). It is still a proposal for renaming FBSS [3]. The text (not necessarilly in the title) has to be revised so it becomes evident that this is a proposal and not a fact!!! Numerous suggestions have previously been brought forward in order to replace the term FBSS, but none has been widely adopted yet!
- Page 2, line 64. The pain after spine surgery can also be remaining without improvement and does not need to be new ("result in the development of post-operative chronic pain").
- The review by Rigoard et al from 2019 in Pain Research and Management (Article ID 3126464) stresses the importance of appropriate psychosocial assessment (including work status) in the evaluation of patients with FBSS and should be added to the reference list.
- Why did the authors choose a numeric value greater or equal to 4 as an avarage global pain score for study inclusion (page 3, second paragraph, line 10) instead of 5 which is the generally used value?
- The colors indicated as beige in Fig 1 and 2 seem more to be pink to me!! Why did the authors call it beige?
- In fig 1 it is stated that the FABQ-W variable is not statistically significant (p<0.091), whereas it in the first paragraph of the Discussion (page 8, first paragraph, line 30) is stated that the FABQ-W score was statistically more likely to be SGH- than SGH+. Why this discrepancy??
- The statement in the second paragraph of the Discussion (page 8, line 32) stating that "a man must rely on his body, to be strong, able to overcome pain and spare no effort" should be omitted! It should be sufficient to say that men traditionally have a more physically demanding work which could explain some of the findings made by the authors.
Author Response
Reviewer comment |
Author’s response |
Manuscript revision (new wording is underlined) |
Reviewer #1 |
|
|
Major points |
|
|
This is a well performed and scientifically stringent study showing a relationship between socioeconomic position and health status in patients with persitent low back and/or leg pain and impaired functional level who have undergone previous spinal surgery. The statistical analysis is solid and the authors conclude that the model that they present using The Social Gradient of Health (SGH) concept may provide new opportunities to improve the medical care pathway of patients with refractory pain after spine surgery. |
We sincerely thank you for your encouragement and relevant comments which definitely improve this manuscript.
|
|
Persistent Spinal Pain Syndrome type 2 is NOT a generally accepted term to replace the acronym FBSS (Failed Back Surgery Syndrome). |
Thank you for pointing out this important issue; we are totally in accordance with your comment. Indeed, Failed Back Surgery Syndrome (FBSS) is still mainly used to define the population and it is also the most heavily criticized due to the pejorative connotation of FBSS suggesting failure or blame (Al Kaisy et al. 2015, PMID: 25595592). Given this context, we had chosen to use the new proposal of Christelis et al. (2021, PMID: 33779730), Persistent Spinal Pain Syndrome Type 2 (PSPS-T2). The manuscript has been updated to provide that FBSS is the main widely term used to describe the population, and to introduce the new proposal of PSPS-T2. In accordance with your comment, this approach nuances our choice to use PSPS-T2 naming. |
Page 1, line 40-42: Failed Back Surgery Syndrome or Persistent Spinal Pain Syndrome type 2 (PSPS-T2) represents one of its most fascinating and disabling conditions.
Page 2, line 57-60: Between 10 % and 50 % of patients who have undergone lumbar spinal surgery still experience the intense persistent pain and impaired function known as Failed Back Surgery Syndrome (FBSS) and which has recently been proposed as Persistent Spinal Pain Syndrome type 2 (PSPS-T2) [1–3]. |
Page 2, line 64. The pain after spine surgery can also be remaining without improvement and does not need to be new ("result in the development of post-operative chronic pain"). |
Thank you for this comment. We agree with your comment which reflects what we want to point out in this sentence. “Even when anatomically successful” evoked that radiological outcomes can be achieved without any pain improvement. To avoid any confusion, the manuscript has been updated accordingly. |
Page 2, line 65-67: Unfortunately, even when anatomically and radiological successful outcomes, this spine surgery may not relieve LBP and/or can result in the development of post-operative chronic pain. |
The review by Rigoard et al from 2019 in Pain Research and Management (Article ID 3126464) stresses the importance of appropriate psychosocial assessment (including work status) in the evaluation of patients with FBSS and should be added to the reference list.
|
Thank you for this relevant proposal. The reference has been now added.
|
Page 2, line 59-61: While PSPS-T2 may affect the mainstream population and has been considered as biologically heterogeneous [5], its development and evolution may also be influenced by psychological and social factors [6–8].
Page 2, line 73-74: In the literature, however, social factors have been shown to affect chronic pain diseases[6,11], particularly throughout the Social Gradient of Health (SGH) examination.
References: 6. Rigoard, P.; Gatzinsky, K.; Deneuville, J.-P.; Duyvendak, W.; Naiditch, N.; Van Buyten, J.-P.; Eldabe, S. Optimizing the Management and Outcomes of Failed Back Surgery Syndrome: A Consensus Statement on Definition and Outlines for Patient Assessment. Pain Res Manag 2019, 3126464, doi:10.1155/2019/3126464. |
Why did the authors choose a numeric value greater or equal to 4 as an avarage global pain score for study inclusion (page 3, second paragraph, line 10) instead of 5 which is the generally used value? |
First, we agree that a cut-off of 5 is generally used, notably in interventional studies. In accordance with the French Health Authorities, a patient is diagnosed as painful with a score of 4 or higher. Similarly, Boonstra et al. (2014, Pain, PMID: 25239073) indicated a cut-off of 3.5 over 10 to consider patient as painful. Given this context, we choose a cut-off of 4 to consider painful patient in our real life observational study. To give you an overview of our patients’ pain score, 3 over 187 had a VAS global score of 4.
The reference of Boonstra et al. (2014, Pain, PMID: 25239073) has been added in the methods section. |
Page 3, line 109 - 111 To be eligible, patients had at least one spinal surgery, post-operative leg and/or LBP for at least six months, and an average global pain score greater than or equal to 4 on the Numeric Pain Rating Scale (NPRS) [18].
References: 18. Boonstra, A.M.; Schiphorst Preuper, H.R.; Balk, G.A.; Stewart, R.E. Cut-off Points for Mild, Moderate, and Severe Pain on the Visual Analogue Scale for Pain in Patients with Chronic Musculoskeletal Pain. Pain 2014, 155, 2545–2550, doi:10.1016/j.pain.2014.09.014. |
The colors indicated as beige in Fig 1 and 2 seem more to be pink to me!! Why did the authors call it beige? |
We have replaced beige by pink. |
Page 11, line 304-305: Variable coloured in pink is not statistically significant (p=0.091)
Page 11, line 368-371: The different shades of pink (from pink to red) make it possible to highlight the consequences of the accumulation of social inequalities associated with SGH-. |
In fig 1 it is stated that the FABQ-W variable is not statistically significant (p<0.091), whereas it in the first paragraph of the Discussion (page 8, first paragraph, line 30) is stated that the FABQ-W score was statistically more likely to be SGH- than SGH+. Why this discrepancy?? |
Thank you for your comment. FABQ-W was indeed not statistically significant in multivariate analysis, while it was in unitivariate analysis. To avoid any confusion, this point has been clarified for FABQ-W and other scores.
|
Page 9, line 312-314: Furthermore, patients with high CSQ-Catastrophizing (in univariate and multivariate analysis) score and FABQ-Work score (in univariate analysis only) were significantly more likely to be SGH- than SGH+. |
The statement in the second paragraph of the Discussion (page 8, line 32) stating that "a man must rely on his body, to be strong, able to overcome pain and spare no effort" should be omitted! It should be sufficient to say that men traditionally have a more physically demanding work which could explain some of the findings made by the authors. |
We thank you for the suggestion. Manuscript updated. |
Page 9, line 326-328: Furthermore, in accordance with the traditional model of masculinity, particularly present among SGH-, men traditionally have a more physically demanding work. |
Reviewer 2 Report
Thank you for allowing me the opportunity to review this manuscript. As a medical Sociologist, I find this research very interesting and very important. Understanding the importance of socioeconomic status and social standing is vital to understanding health outcomes.
The manuscript is in great shape, and I only have few suggestions in that regard:
Abstract
Line 47: “SHG” should be SGH (the first acronym in that sentence, the second one is correct)
Introduction
Line 56: I would suggest using the “%” sign next to both “10” and “50” or using the word “percent” instead, unless this goes against the formatting style used for this article
Materials and Methods
Line 113: Shouldn’t “and” be “or” instead?
Multivariate Analysis
Lines 283-287: I assume females were coded as “0” since the coefficient has a negative sign? If so, you may want to state that in the text, since males are usually coded “0” (the reference category). The reason I assume females were coded “0” is because you state they are more likely to be SGH+ patients, this is with a -0.349 coefficient.
Figure 2
I’m not understanding the message this figure it trying/supposed to convey. It would be nice to give more detail on this figure. For instance, why is the green area (high SGH) in the middle with the other colors on the outer part of the pyramid?
Line 367: Should read “Thomas et al.”
Author Response
We sincerely thank you for your positive feedback; we appreciate your relevant comments which have helped us to improve this manuscript.
Reviewer comment |
Author’s response |
Manuscript revision (new wording is underlined) |
Reviewer #2 |
|
|
Major points |
|
|
Thank you for allowing me the opportunity to review this manuscript. As a medical Sociologist, I find this research very interesting and very important. Understanding the importance of socioeconomic status and social standing is vital to understanding health outcomes.
|
We sincerely thank you for your positive feedback; we appreciate your relevant comments which have helped us to improve this manuscript.
|
|
Line 47: “SHG” should be SGH (the first acronym in that sentence, the second one is correct) |
Thank you for pointing out this mistake. SGH has been now well spelled. |
Page 01, line 47-48: 85.3% of PSPS-T2 patients in our study had low SGH. |
Line 56: I would suggest using the “%” sign next to both “10” and “50” or using the word “percent” instead, unless this goes against the formatting style used for this article |
Thank you for this suggestion. “%” sign has been now added after both “10” and “50”. |
Page 02, line 57-60: Between 10 % and 50 % of patients who have undergone lumbar spinal surgery still experience the intense persistent pain and impaired function known as Failed Back Surgery Syndrome (FBSS) and which has recently been proposed as Persistent Spinal Pain Syndrome type 2 (PSPS-T2) [1–3] |
Line 113: Shouldn’t “and” be “or” instead? |
Thank you for point out this mistake. Manuscript updated. |
Page 03, line 113-118: Non-inclusion criteria: Patient is or has been treated with Spinal Cord Stimulation, subcutaneous or peripheral nerve stimulation, an intrathecal drug delivery system; has life expectancy of less than 12 months beyond study enrollment; Patient is unable to undergo study assessments or to complete questionnaires independently; is a member of a vulnerable population; or investigator suspects substance abuse that might confound the study results. |
Lines 283-287: I assume females were coded as “0” since the coefficient has a negative sign? If so, you may want to state that in the text, since males are usually coded “0” (the reference category). The reason I assume females were coded “0” is because you state they are more likely to be SGH+ patients, this is with a -0.349 coefficient. |
Thank you for highlighting this point. We changed the coding of the gender categories to 0 for men and 1 for women, which, as expected, reversed the sign of the regression coefficient. Manuscript updated. |
Page 08, line 290-291: Women were more likely to be SGH+ than SGH- (coef.=0.349; IC95% [0.070;0.627;]; p=0.014]). |
I’m not understanding the message this figure it trying/supposed to convey. It would be nice to give more detail on this figure. For instance, why is the green area (high SGH) in the middle with the other colors on the outer part of the pyramid? |
Thank you for this comment. Since this figure is crucial to understand our hypothesis, your feedback allows us to clarify its explanation. The main message of this figure was to illustrate the cumulative influence of SGH on the theoretical proportion of patients according to their position in the socioeconomic hierarchy depending on the PSPS-T2 patients’ illness trajectory. This figure shows that (i) the proportion of SGH+ decrease while the pain illness level increase (from acute pain to PSPS-T2); (ii) the proportion of SGH- increase while the pain illness level increase. We have chosen to keep green area (representing SGH+) in the middle to show the decrease of SGH+ proportion associated with the increase of illness level. In this context, we can ensure to show SGH+ in every illness level that it was more complicated with SGH+ on side.
Please let us know if the explanation of the figure is now clear. |
Page 11, line 368-371: This figure illustrates the cumulative influence of the SGH on the theoretical proportion of patients according to their position in the socioeconomic hierarchy at each stage of the PSPS-T2 patients’ illness trajectory. The different shades of pink (from pink to red) indicated the new proportion of SGH- patient at each pain illness level from acute to refractory PSPS-T2. This figure shows that (i) the proportion of SGH+ decrease while the pain illness level increase (from acute pain to PSPS-T2); (ii) the proportion of SGH- increase while the pain illness level increase.
|
Line 367: Should read “Thomas et al.” |
Thank you for pointing out this mistake. Manuscript updated. |
Page 11, line 376-378: Thomas et al. [42] found that functional disability was positively correlated with catastrophizing and kinesiophobia among 50 patients with chronic LBP. |
This manuscript is a resubmission of an earlier submission. The following is a list of the peer review reports and author responses from that submission.